# Pairwise frictional profile between particles determines discontinuous shear thickening transition in non-colloidal suspensions

Jean Comtet[1], Guillaume Chatté[2], Antoine Niguès[1], Lydéric Bocquet[1], Alessandro Siria[1] & Annie Colin[2,3]

The process by which sheared suspensions go through a dramatic change in viscosity is known as discontinuous shear thickening. Although well-characterized on the macroscale, the microscopic mechanisms at play in this transition are still poorly understood. Here, by developing new experimental procedures based on quartz-tuning fork atomic force microscopy, we measure the pairwise frictional profile between approaching pairs of polyvinyl chloride and cornstarch particles in solvent. We report a clear transition from a low-friction regime, where pairs of particles support a finite normal load, while interacting purely hydrodynamically, to a high-friction regime characterized by hard repulsive contact between the particles and sliding friction. Critically, we show that the normal stress needed to enter the frictional regime at nanoscale matches the critical stress at which shear thickening occurs for macroscopic suspensions. Our experiments bridge nano and macroscales and provide long needed demonstration of the role of frictional forces in discontinuous shear thickening.

[1] Laboratoire de Physique Statistique, Ecole Normale Supérieure, UMR CNRS 8550, PSL Research University, 24 rue Lhomond, 75005 Paris Cedex 05, France. [2] ESPCI Paris, Sciences et Ingénierie de la Matière Molle, CNRS UMR 7615 10, PSL Research University, rue Vauquelin, F-75231 Paris Cedex 05, France. [3] Université de Bordeaux, Centre de Recherche Paul Pascal 115 avenue Schweitzer, 33600 Pessac, France. Correspondence and requests for materials should be addressed to A.C. (email: annie.colin@espci.fr).

Dense suspensions of solid particles immersed in a Newtonian solvent display complex flow properties. Shear thickening, where the resistance to flow increases when the dispersion is stirred is one of the most intriguing behaviour[1,2]. In the extreme situation of discontinuous shear thickening, shear viscosity increases by orders of magnitude at a given shear rate. Under impact, the formation of a dynamic jamming front makes the fluid so resistant that a person can can run on it[3–5]. This phenomenon, well documented[6,7] yet not really understood, is essential from the perspective of applications and materials. Flows of concentrated dispersions are ubiquitous in nature and industry: water or oil saturated sediments, muds, crystal-bearing magma[8], concrete, silica suspensions, cornflour mixtures, latex suspensions and clays are examples of shear thickening systems. Industrially, these systems can have disastrous effects by damaging mixer blades or clogging pipes. Discontinuous shear thickening may also be harnessed and desirable when engineering composite materials, for shock-absorbing materials or soft body armour[9].

Despite an extensive characterization of discontinuous shear-thickening transitions at the macroscale, there is still no clear understanding of the microscopic mechanisms at play in this transition, principally owing to the challenges associated with quantitative frictional measurements at the nanoscale[10], especially for pairs of particle[11].

A long-standing view is that thickening is driven by hydrodynamic and Brownian forces[12]. At large shear rate, these forces create highly dissipative transient clusters of particles, due to the singular lubrication flows between the particles. When normal elastohydrodynamic contact forces are taken into account (without solid friction), these simulations capture continuous shear thickening and large increase in suspension viscosity[13]. However, this model predicts shear rate-independent rheology for non-Brownian systems and broader transition that observed experimentally.

To get around this issue, recent works[14,15] propose a new picture that neglects thermal fluctuations and put forward the role of (nanoscale) repulsive and frictional forces. At low pressure, neighbouring particles are separated by a gap filled with solvent and interact via hydrodynamic forces. At high pressures, repulsive forces are overcome, leading to frictional contacts and shear thickening.

At this stage, this picture and the role played by frictional forces have been validated through numerical simulations but only indirectly through experiments at the level of the suspension[2,6,16].

In the present work, we bridge this gap between nano and macro-scales. Taking advantage of quartz-tuning fork based atomic force microscope with subnanonewton resolution, we build upon state of the art procedures[17,18] to measure the pairwise force profile and the frictional interactions between pairs of particles and correlate these measurements to the macroscale rheology of suspensions. We study two well-known shear-thickening systems: polyvinyl chloride (PVC) suspended in various solvents[19,20] and cornstarch particles suspended in water[21,22]. Our measurements provide a clear view of discontinuous shear-thickening transition as the breakdown of lubricated contact between particles, at a critical normal force.

## Results

**Experimental set-up**. We present in Fig. 1a the schematic of the experimental set-up. Briefly, we glue an electrochemically etched tungsten tip of ∼50 nm end radius to a millimetric quartz tuning fork, which serves as our force sensor. Using an in-house-built nano-manipulator in a scanning electron microscope (SEM), we then glue one individual particle to the end of the tungsten tip (Fig. 1b).

During a typical experiment, the attached particle is immersed in solvent and brought into contact to another bead fixed on the substrate, while monitoring the force profile (Fig. 1a). Further characterization of substrate roughness and topography are shown in Supplementary Figs 1 and 2.

To measure simultaneously normal and tangential force profiles between the two approaching particles, we simultaneously excite the tuning fork via the piezo-dither at two distinct resonance frequencies $f_N \approx 31$ kHz and $f_T \approx 17$ kHz, corresponding to the excitation of both normal (blue arrows) and shear modes (red arrows) of the tuning fork, as shown in Fig. 1c. Both modes correspond to symmetric excitation of the prongs, leading to negligible motion of the centre of mass and high-quality factor of the oscillator[23]. Monitoring changes in the resonance of each modes (Fig. 1d) allows us to measure respectively the normal and tangential force profile between the two objects.

We now detail how to obtain force profile for each normal ($N$) and tangential ($T$) modes (Fig. 1d). When excited close to a resonance frequency, the system can be modelled as a mass-spring resonator[24], whose resonance profile will be modified by the interacting forces. First, position dependent forces $\mathbf{F}(x, z)$ will lead to shifts in the resonance frequency of the oscillator (comparing black and red resonance curves in Fig. 1d). Hence, the spatial force gradient $\nabla\mathbf{F}$ (N m$^{-1}$) is proportional to the shift in the resonance frequency $\delta f^i$ (Hz) according to[24]:

$$\nabla F_i = \nabla\mathbf{F}.\mathbf{e}_i = -2k_{TF}^i \left(\frac{\delta f^i}{f_0^i}\right) \text{ with } i \in \{N, T\}, \quad (1)$$

where $f_0^i$ (Hz) and $k_{TF}^i$ (N m$^{-1}$) are the spring constants, respectively, associated with normal ($N$) and tangential ($T$) oscillation modes, $\mathbf{e}_i$ corresponds to the unit vector along direction $i$.

We can then express the frictional forces applied on the bead as the sum of viscous-like friction force (proportional to speed) and solid friction force (independent of speed and corresponding to sliding friction for tangential motion, $i = T$):

$$F_D^i = \gamma_i \cdot v_i + F_s^i \cdot \frac{v_i}{\|v_i\|} \text{ with } i \in \{N, T\}, \quad (2)$$

where $v$ (m s$^{-1}$) is the oscillating speed, $\gamma$ (kg s$^{-1}$) is a viscous-like friction coefficient and the second term $F_s$ [N] characterizes the solid friction force, only dependent of the sliding direction $v_i/\|v_i\|$. The friction forces $F_s$ are assumed to be velocity independent as experimentally confirmed a posteriori. Over one oscillation period, all friction forces in equation (2) have zero mean, only leading to changes in the height and broadening of the resonance peak (comparing black and red resonance in Fig. 1d). It is noteworthy that we can not measure static friction forces with our set-up. However, we expect static friction to appear concurrently to sliding dynamic friction.

Keeping a constant oscillation amplitude $a_i$ at resonance requires the external excitation force $F_{ext}$ (due to the piezo dither) to be equal to the to the sum of all dissipative forces $F_D$. As the external force $F_{ext}$ is directly proportional to the excitation voltage $E_{ext}$, the sum of all dissipative forces $F_D$ is directly measured by tracking the excitation voltage necessary to keep a constant oscillation amplitude (see Methods for further details, Supplementary Figs 4–6 and Supplementary Note 1).

**Nanoscale force profile**. We show in Fig. 2 the typical force profile measured between two approaching PVC beads in good solvent. Monitoring changes in the resonance (Fig. 1d) for the two oscillating modes (Fig. 1c) allows us to characterize pairwise

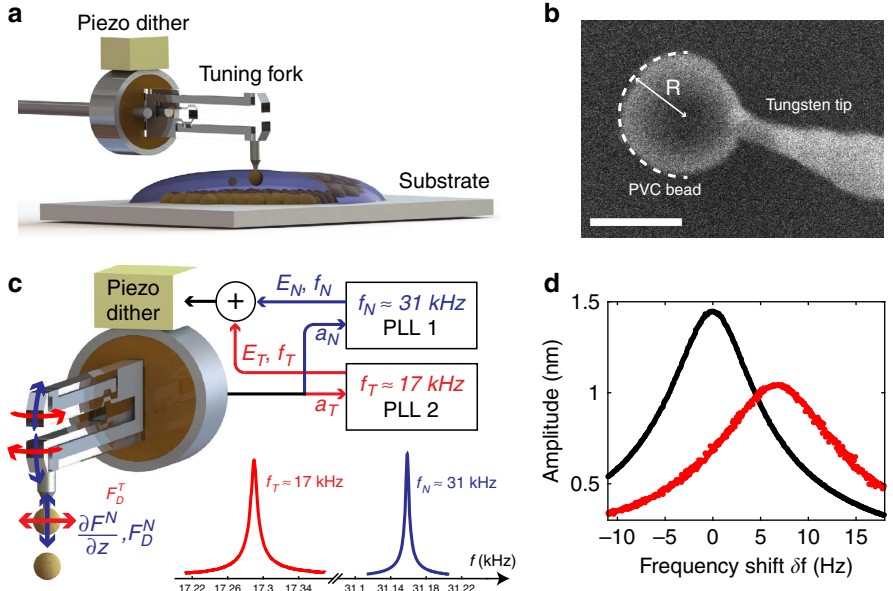

**Figure 1 | Experimental set-up.** (**a**) Schematic of the set-up. A particle is glued to the quartz-tuning fork based AFM, immersed in the liquid and approached to the substrate, made of casted particles. (**b**) SEM image of a 0.6 μm radius PVC bead glued to the tungsten tip. Scale bar, 1 μm. (**c**) During the experiment, the tuning fork is excited at two distinct frequencies, corresponding to mechanical oscillation of both normal (*N*, blue) and tangential/shear (*T*, red) modes. Two phase-locked loops (PLL) track the frequency of the resonance peaks, allowing characterization of normal and tangential force gradients $\nabla \mathbf{F}$ (equation (1)) and dissipative frictional forces $\mathbf{F}_D$ (equation (2)). (**d**) Typical resonance curve of the normal mode, for a bead in liquid (black) and in contact to another bead on the substrate (red).

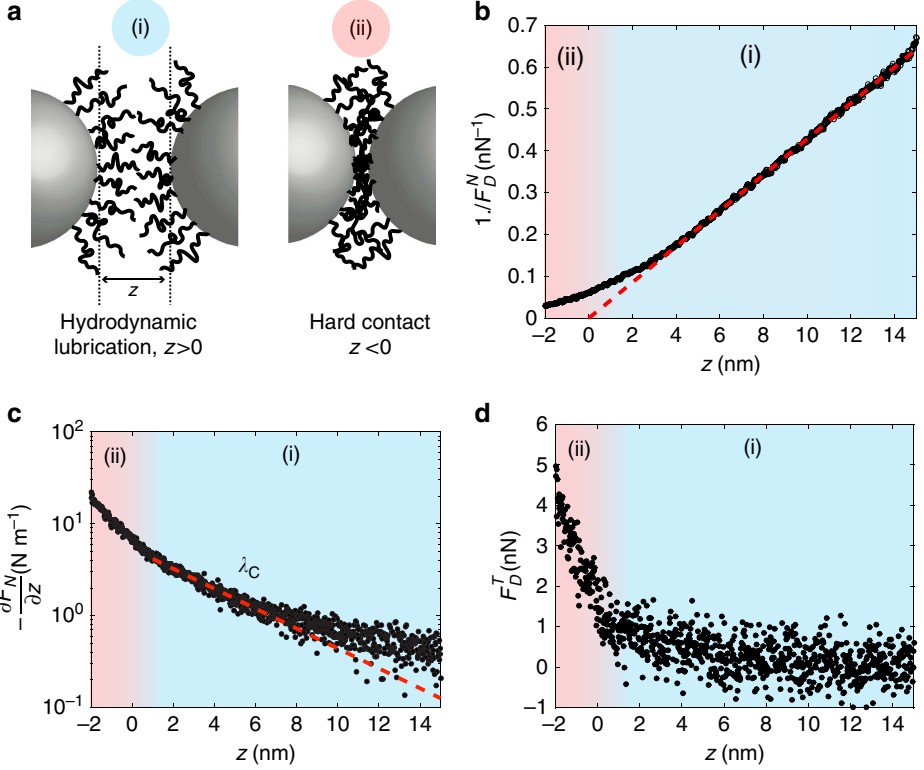

**Figure 2 | Characterization of nanoscale force profile.** (**a**) When immersed in Dinch a good solvent, polymer brush form at the surface of PVC particles. Distance between bead's no shear plane is written *z*, where $z = 0$ correspond to hard contact. (*i*) $z > 0$ Entropic repulsion between polymer brushes. (*ii*) $z < 0$ hard contact. (**b**) Inverse of the normal friction force $1/F_D^N$. (**c**) Projection of the normal force gradient along the normal direction $-\partial F_N/\partial z$. (**d**) Tangential friction force $F_D^T$. The radius of the attached bead is 0.6 μm and beads are immersed in pure Dinch.

interparticles interactions through the normal dissipation $F_D^N$ (Fig. 2b), the projection of the normal force gradient along the normal direction—$\partial F_N/\partial z$ (Fig. 2c) and the tangential friction force $F_D^T$ (Fig. 2d).

We first show in Fig. 2b the inverse of the normal dissipative force $1/F_D^N$. The red line is a linear fit of the inverse dissipation, showing that normal dissipation $F_D^N$ is characterized by hydrodynamic drainage and Stokes law as the beads are separated from each other:

$$F_D^N \sim \frac{\eta R^2 v}{z}, \qquad (3)$$

where $\eta \approx 40\,\mathrm{mPa\,s^{-1}}$ is solvent viscosity (see Supplementary Section S2.3), $v = a\omega$ $(\mathrm{m\,s^{-1}})$ is the typical speed of the oscillating particle, $z$ (m) is the distance between the two no-shear planes and $R$ (m) is an equivalent bead radius. The intersection of the red line with the horizontal axis defines the hydrodynamic zero ($z = 0$), which defines the absolute position of the no-shear planes between the two objects (vertical dotted lines Fig. 2a). This zero defines two domains, corresponding to (i) $z > 0$, hydrodynamic lubrication (light blue) and (ii) $z < 0$ hard contact (light red). It is noteworthy that for confinement below $\approx$ 3 nm, we observe a deviation from Stokes hydrodynamics with a regularization of the hydrodynamic divergence, possibly stemming from elastohydrodynamic interactions[13,25,26]. The dissipative normal forces measured for $z < 0$ may be due to viscoelasticity of the PVC particles.

We now turn to the normal force gradient $-\partial F_N/\partial z$ $(\mathrm{N\,m^{-1}})$ shown in Fig. 2c. For the two approaching particles, we observe an increasing repulsive normal force gradient ($-\partial F_N/\partial z > 0$) before contact between the two particles ($z > 0$, blue zone). These repulsive forces vary steadily and smoothly with distance, whereas normal dissipation is dominated by hydrodynamics during the approach (Fig. 2b). We thus interpret these repulsive forces as a signature of the entropic repulsion between polymer brushes forming at the surface of the PVC beads, due to the effect of the plasticizing solvent[19,27] (Fig. 2a). We can characterize the steepness of this repulsive profile right before contact by an exponential-like law $F \approx \exp(-z/\lambda)$ with $\lambda \approx 4\,\mathrm{nm}$ (Fig. 2c, red dotted line)[28]. Upon contact, the steepness of the repulsive profile increases slightly.

We show in Fig. 2d the tangential dissipative friction force $F_D^T$ (tangential mode $T$, Fig. 2c). Before contact (blue zone (i)), tangential forces are below 1 nN, consistently smaller that the normal hydrodynamic dissipative forces which are of the order of 5–10 nN. Upon contact (red zone, (ii)), we observe a clear increase of frictional forces. We note that, depending on the respective surface states of the beads, contact can also occur before the hydrodynamic zero (for $z > 0$), due to the presence of asperities on one of the bead surface (see Supplementary Fig. 5). Finally, we note that the fact that we recover solvent viscosity $\eta$ in the dissipative normal force and that there is low tangential lubrication forces before contact are a signature of the absence of brush interpenetration in the probed experimental conditions[29].

**Critical load friction**. We now turn in Fig. 3 to the nature of the frictional profile, as uncovered by the two regimes shown in Fig. 2. We plot in Fig. 3a the typical form of the tangential dissipative force $F_D^T$ as a function of the normal load $F_N$, obtained by integrating the normal force gradient[30]. In the first regime of hydrodynamic lubrication (blue zone, (i)), tangential frictional forces are small and arise purely from hydrodynamic interactions, whereas a normal load $F_N$ can be sustained due to entropic repulsion of the brushes. This situation results in a friction

coefficient as low as $\mu \approx 0.02$, as observed in previous friction studies on polymer brushes in SFA[29,31–34].

On a critical normal load $F_N^C$ corresponding to the force necessary to completely compress the polymer layers and reach hard contact, the system switches to a second state characterized by a sharp increase in friction (Fig. 2d, red zone (ii)). This second regime is well characterized by Amontons–Coulomb laws, with a proportionality between tangential frictional forces and normal load: $F_D^T = \mu(F_N - F_N^C) + F_V^C$, where $F_V^C$ is the tangential viscous dissipation right before contact. Moreover, as shown in Fig. 3b, the friction coefficient between two beads is independent of the sliding speed for tangential speeds above $200\,\mathrm{\mu m\,s^{-1}}$, a clear characteristic of solid-like friction (relative speed is changed here through the oscillation amplitude $a_0$). The independence of sliding friction with speed validates a posteriori our choice for the form of equation (2).

Finally, we show in Fig. 3c the distribution of friction coefficient obtained over 30 different pairs of beads. As characterized in Fig. 3b, the friction coefficient is a well-defined property of each particle interactions (Fig. 3b) but also depends on the local physicochemical, geometrical, mechanical and roughness surface state of the two sliding beads. We find a mean interparticle friction coefficient $\mu = 0.45 \pm 0.2$, in very good agreement with the macroscopic friction coefficient of PVC on PVC[35].

## Discussion

We now come back to the macroscale behavior of non-Brownian suspensions.

As shown in Fig. 4a,b, we measured using standard rheometry the flow curves (that is, the relation between the applied shear stress $\sigma$ and the measured shear rate $\dot\gamma$) for various solutions of PVC particles in mixtures of Dinch and mineral oil[19,20], and cornflour particles in water[21,22] at various solid fractions (see Supplementary Fig. 3 for details). All these suspensions exhibit a discontinuous shear thickening transition above a critical shear stress $\sigma_C$ (defined as the critical shear stress above which the viscosity starts to increase as a function of the shear rate[2]). As found previously, $\sigma_C$ (contrary to the corresponding shear rate $\dot\gamma_C$) does not depend upon the solid fraction of particles $\phi^2$ for high solid fractions (Fig. 4b).

Following recent models, $\sigma_C$ corresponds to the shear stress required to obtain a particle normal stress high enough to overcome the repulsive forces and to transit from lubricated to frictional contacts[15,36]. If the shear-thickened state is characterized by frictional interactions between the particles, there should be a correlation between the macroscale critical shear stress $\sigma_C$ and the critical load $F_N^C$ uncovered in Fig. 3. To verify this correlation, we measured the microscopic critical load $F_N^C$ for each of the systems of Fig. 4a (typical distribution of normal critical force is shown in Supplementary Fig. 7).

We report in Fig. 4b the macroscale shear stress $\sigma_C$, versus the nanoscale critical force $F_N^C$ for each discontinuous shear-thickening systems. For both PVC and cornstarch systems, we see a clear correlation between those two nanometric and macroscopic quantities, with the critical stress at the macroscale varying proportionally to the critical force needed to enter into frictional contact at the nanoscale:

$$\sigma_C = \beta \cdot \frac{F_N^C}{\pi R^2}. \qquad (4)$$

We find here a proportionnality coefficient $\beta_{PVC} \approx 0.006$ and $\beta_{Cornstarch} \approx 0.02$, in relatively good agreement with predictions from simulations performed on smooth particles ($\beta_{simu} \approx 0.05$ for a friction coefficient $\mu = 1$ (ref. 36)). Those coefficients characterize stress transmission from the suspension to

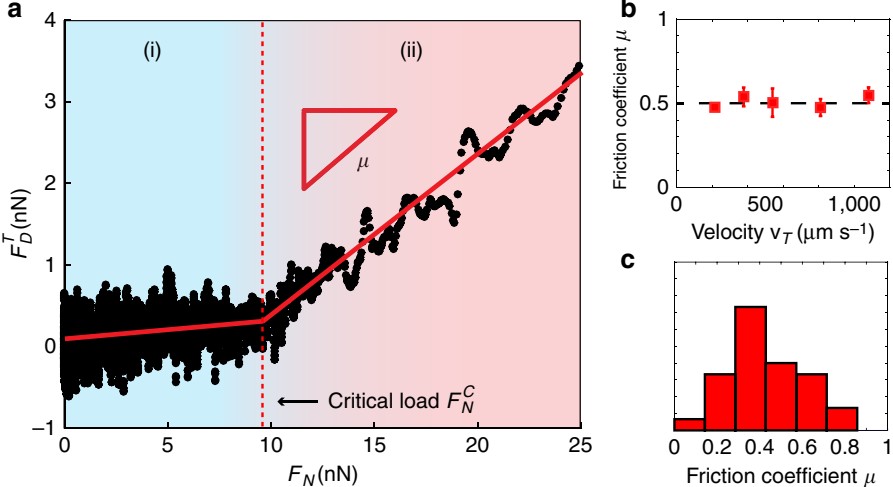

**Figure 3 | Critical load frictional profile.** (**a**) Tangential friction forces versus normal load, showing a transition between (i) a hydrodynamically lubricated low friction regime, to (ii) a solid-like high friction regime. (**b**) Variation of the friction coefficient $\mu$ with sliding velocity $v$ for one pair of beads. Error bars are SEM for $N > 2$. (**c**) Distribution of the friction coefficient found on 30 different pairs of beads. The radius of the attached bead is 0.5 μm and the solvent between the two beads is pure Dinch (Methods).

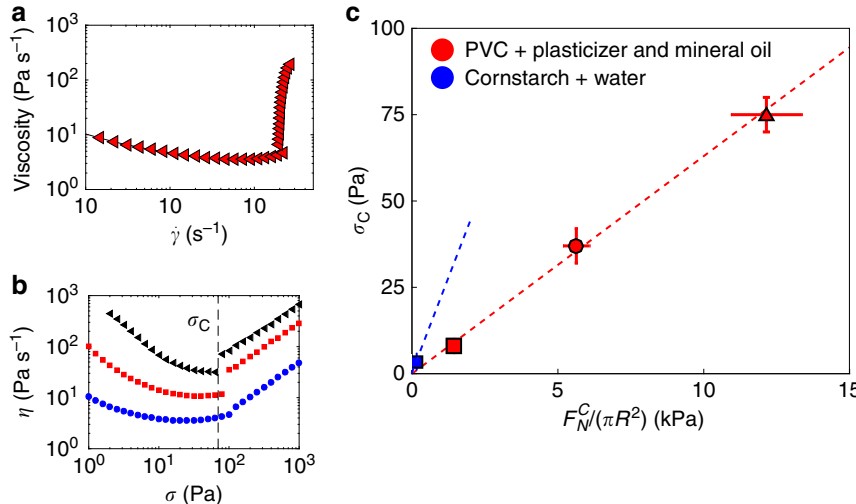

**Figure 4 | Nanoscale critical load determines macroscale critical shear stress.** (**a**) Shear viscosity as a function of the shear rate for suspensions of PVC in pure Dinch and a volumic solid fraction equal to 60%. (**b**) Flow curves for PVC in pure Dinch, for volumic solid fractions equal to 66%, 64% and 60% (from top to bottom) allowing the characterization of the critical shear stress $\sigma_C$. The vertical dashed line corresponds to the critical shear stress $\sigma_C = 75$ Pa $\pm 5$. (**c**) Correlation between the critical normal stress $F_N^C$ and critical shear stress $\sigma_C$ for PVC in 100% (red triangle), 90% (red circle) and 67% (red square) plasticizer and cornstarch in water (blue square). Horizontal error bars are SEM for PVC, with $N > 15$ and s.d. for Cornstarch ($N = 4$). Red and blue dotted lines are respectively linear fit to the PVC and cornstarch systems.

the particles level and depend on the microscopic friction coefficient[36] for both static and sliding friction, as well as particle shape and roughness. It is noteworthy that in the simulations, the values of the static and dynamic friction are assumed to be the same, which may not be the case in our situation. Macroscopic roughness may also block the particles and affect the value of $\beta$ found by numerical simulations.

Let us underline that $F_N^C$ does not depend upon the relative tangential velocities between particles in the range of experimental data. Moreover, both $F_N^C$ and $\tau_C$ are measured for approximatively the same range of relative velocities between the two particles. In the macroscopic experiments, the relative velocity between two particles can be approximated as $v \approx \dot{\gamma}_C R$ at the onset of the shear thickening transition and varies between 4 and 200 μm s$^{-1}$. In the AFM experiments, the normal root

mean square (RMS) speed is ~30–200 μm s$^{-1}$ and the tangential RMS speed 150–800 μm s$^{-1}$.

Our unique experimental set-up allowed for the first time to characterize frictional interactions between pairs of particles from discontinuous shear-thickening suspensions. We explain unambiguously discontinuous shear-thickening transition as stemming from the breakdown of lubrication between particles and the onset of hard frictional contacts. Even though normal elastohydrodynamic forces appear for very small separation distances (Fig. 2b, $z < 2$ nm), they do not contribute to the observed shear-thickening transition. Indeed, we find that the critical shear stress $\sigma_C$ at the shear-thickening transition varies as a function the contact forces (Fig. 4c). This is opposite to predictions of numerical simulations proposed by Jamali et al.[13], for which the shear-thickening transition is found to be

independent of particle elastic modulus if stemming from elastohydrodynamic forces.

Our measurements pave the road for direct simulation of the rheological properties of dispersions. In particular, the distribution of friction coefficients, contact forces and size distribution of the dispersion should be taken into account in future numerical simulations to capture at which solid fraction the dispersion evolves to a continuous or discontinuous shear thickening transition.

Going further, our measurements provide for the first time a firm microscopic description for the rheological behaviour of suspensions of particles. Similar measurements could be extended to the regimes of high shear and high load, and might explain the shear-thinning regimes often measured after the shear-thickening transition. Finally, rheological behaviors as diverse as shear thinning[37], self-filtration[38] or particle migration could be disentangled and better understood through similar approaches.

## Methods

**PVC suspensions, plasticizer and mineral oil.** The mean particle radius of PVC particles, defined as $R_{32} = \langle R^3 \rangle / \langle R^2 \rangle$ is 1 μm. The size distribution is log-normal and the s.d. estimated using the volume distribution is 45%. We measured a RMS roughness of 2.2 nm for PVC particles.

As a plasticizer for PVC particles, we use 1,2-cyclohexane dicarboxylic acid di-isononyl ester (Dinch) supplied by BASF. This organic liquid Dinch slightly dissolves the outer part of the particles and creates a polymer brush around them. This brush enables stabilization of the suspension due to steric repulsion[19]. At high temperature ($T \geq 100\,°C$), Dinch can dissolve the PVC particles[39]. At room temperature, this process is much slower and takes > 1 year.

Degree of plasticization can be changed by using a mix of mineral oil and plasticizer[20], because mineral oil has a low affinity with PVC. Mineral oil viscosity standards were provided by Paragon Scientific Ltd. Same viscosity as Dinch (41.1 mPa s$^{-1}$ at 25 °C) was achieved by mixing two different viscosity standards (55.7 and 29.0 mPa s$^{-1}$). Resulting viscosity was checked in a shear rate range from 1 to 100 s$^{-1}$ for different temperatures. Same viscosity at 25 °C and same temperature dependence of viscosity (range 20–25 °C) were found between plasticizer and obtained mineral oil.

Concerning PVC, we report results (both AFM and rheological experiments) for three different plasticizing liquids: (1) 100 vol.% plasticizer, (2) 90 vol.% plasticizer + 10 vol.% mineral oil, and (3) 67 vol.% plasticizer and 33 vol.% mineral oil.

**PVC substrate.** To make the substrate, a given amount of PVC powder is introduced into a metallic mold laying on a glass slide. A counter-mould is used on top. The mould and counter-mould are then transferred into a hot press and compressed 5 min at 150 °C and 20 bars. Then, the sample is cooled down, resulting in a compact and transparent piece of PVC. Even if the pressing temperature is higher than the PVC glass transition temperature ($T_g = 80\,°C$), the original shape and surface topography of the particles is preserved (see Supplementary Fig. 1).

**Cornstarch.** Cornstarch was supplied by Sigma-Aldrich and used without further modification. It contains ~73% amylopectin and 27% amylose with particle diameter around 14 μm (polydispersity 40% from static light scattering)[22]. Owing to very low volume used and evaporation-related problems, AFM measurements for cornstarch were done in pure water as a suspending liquid. Rheological measurements were also carried out in pure water for the cornstarch suspensions. Substrates were made by gluing cornstarch particles on a flat silicon substrate using cyanoacrylate glue (see Supplementary Fig. 2). We measured a RMS roughness of 14 nm for the cornstarch particles.

**Particle gluing under SEM and particle size.** PVC and cornstarch particles are glued to the etched tungsten tip of the tuning fork using SEMGLU from Kleindiek and a nanomanipulation station in-situ SEM (FEI Nova NanoSEM 450) (see Fig. 1b). Measurements presented in the supplemental materials and main text were obtained for four different attached particles (radius of 0.6 and 0.5 μm for the two PVC particles, and 3.8 and 4 μm for the two cornstarch particles).

**Measurement of frictional and dissipative forces.** We consider below the equation of motion for the tuning fork close to its resonance frequency, in the presence of both viscous and sliding friction. We recall that the friction force $F_D^i$ can be expressed as the sum of a viscous like friction coefficient and a sliding friction force: $F_D^i = \gamma_i \cdot v_i + F_S^i \cdot \frac{v_i}{\|v_i\|}$ with $i \in \{N, T\}$. $\gamma_i \cdot v_i$ is the viscous force, with $\gamma_i$ the viscous friction coefficient and $F_S^i \cdot \frac{v}{\|v\|}$ the sliding friction force (equations (1)

and (2)). In its complex form and for the tangential and the normal directions, the equation of motion becomes:

$$-\omega^2 \underline{X} + i\omega \frac{\gamma}{m} \underline{X} + \omega_0^2 \left(1 - \frac{1}{2k_i} \nabla \mathbf{F} \cdot \mathbf{e_i}\right)^2 \underline{X} = \frac{1}{m}\left(F_{ext}^i - \frac{2F_S^i}{\pi}\right) \quad (5)$$

where $\omega$ is the excitation frequency, $\omega_0 = \sqrt{k_i/m}$ is the natural frequency of the oscillator, with $m$ the equivalent mass and $k_i$ the equivalent stiffness of the mode $i \in \{N, T\}$. $\underline{X}$ is the complex oscillator amplitude, $F_{ext}^i$ is the external piezoelectric excitation force and $F_S^i$ is the friction force. Upon approach, $\gamma$, $F_S^i$ and $\nabla \mathbf{F}$ will vary with position. The factor $2/\pi$ stems from the first Fourier coefficient of the square-like shape of the solid friction force. From equation (5), we obtain a fundamental relation between the amplitude at resonance $a_0$, the quality factor $Q = m\omega_0/\gamma$ characterizing viscous dissipation and the solid friction force $F_S^i$. Neglecting the force gradient compared to the very large stiffness of the tuning fork, we obtain:

$$\frac{a_0}{Q} = \frac{\left(F_{ext}^i - 2F_S^i/\pi\right)}{k_i} \quad (6)$$

The external excitation force necessary to keep a constant oscillation amplitude $a_0$ can then write as the sum: $F_{ext}^i = \gamma a_0 \omega_0 + \frac{2F_S^i}{\pi} = \gamma v + \frac{2F_S^i}{\pi} = F_D^i$. This external force is proportional to the piezoelectric excitation voltage $F_{ext}^i = CV_{ext}$, where $C$ is a factor calibrated at the beginning of each experiment. When the two particles are distant, we get $C = a_0^* k_i / (Q^* V^*)$ from a lorentzian fit of the resonance curve. Monitoring the excitation voltage necessary to keep a constant oscillation amplitude $a_0$ gives us a direct measure of the sum of all forces acting on the tuning fork as $F_{ext}^i = F_D^i = CV_{ext}$ to obtain an amplitude $a_0$ typically equal to 1–50 nm. Both shear and normal modes of the tuning fork can be excited at the same time by summing the two excitation voltage at each frequency (Fig. 1c). Owing to the very large differences in resonance frequency, the two modes are uncoupled. This uncoupling can further be verified by checking that changes in the excitation voltage for one mode do not affect the resonance curve of the second mode. We take $k_N = 40\,kN\,m^{-1}$ for the 32 kHz mode and $k_T = 12\,kN\,m^{-1}$ for the 18 kHz mode. In practice, two phase-locked loops allow us to track the two centre resonance frequencies $f_N$ and $f_T$. A proportional-integral-derivative controller (PID) keeps the oscillation amplitude $a_T$ of the tangential mode constant, allowing a direct measure of the frictional forces by monitoring the amplitude of the excitation voltage $E_T$. A fixed amplitude of the excitation voltage $E_N$ is applied to the normal mode and dissipation is measured by monitoring the oscillation amplitude $a_N$. The electronic lock-in and phase-locked loops s are implemented using a Nanonis from (SPECS Zurich) and a HF2LI Lock-in Amplifier (Zurich Instrument).

**Characterization of PVC suspensions.** We prepare our dispersions by weighting a given amount of PVC particles, a given amount of Dinch and a given amount of mineral oil. The solid fractions are then calculated knowing the density of PVC $\rho_{PVC} = 1.38\,g\,cm^{-3}$, the density of Dinch $\rho_{Dinch} = 0.95\,g\,cm^{-3}$ and the density of mineral oil $\rho_{oil} = 0.84\,g\,cm^{-3}$. Suspensions are stirred 5 min at 1,000 r.p.m. using a Dispermat LC55 (VMA Getzmann), to ensure good dispersion state. Samples were freshly mixed for each experiment. This protocol was found to produce reproducible samples. The solid volume fraction of the suspension is defined as the volume of particles divided by the total volume: $\phi = \frac{m_{PVC}/\rho_{PVC}}{m_{PVC}/\rho_{PVC} + m_{solvent}/\rho_{solvent}}$.

Rheology was measured using a stress-controlled DHR-3 rheometer (TA Instruments) equipped with a smooth cone (diameter $D = 40\,mm$, angle = 2°, truncation gap = 54 μm). A logarithmic stress sweep (50 s per point) between 1 and 1,000 Pa (10 points per decade) was performed to measure the flow curve. Flow curve obtained matches the viscosity found using 'peak holds' of shear rate (flow steady state is reached within 10 s). No wall slip was measured in these experiments. This was checked indirectly by measuring velocity profiles using ultrasounds in Couette cells.

**Characterization of Cornstarch suspensions.** Dispersions were also prepared by weighting a given amount of cornstarch and a given amount of water. The solid fractions are then calculated knowing the density of cornstarch $\rho_{cornstarch} = 1.63\,g\,cm^{-3}$ and the density of water $\rho_{water} = 1.00\,g\,cm^{-3}$. The solid volume fraction of the suspension is also defined as the volume of particles divided by the total volume. Samples were freshly mixed for each experiment. Rheology was measured using a stress controlled DHR-3 rheometer (TA Instruments) equipped with a hatched plate (diameter $D = 40\,mm$) similar to that in previous works[22]. Contrary to PVC suspensions, both lower and upper plate are hatched to avoid wall slip. Flow curves were obtained with a logarithmic stress sweep from 0.1 to 100 Pa (10 points per decade). Each point was measured during 10 s, which was long enough to ensure equilibrium, while avoiding water evaporation and/or particles sedimentation. At high stresses (see Supplementary Fig. 3), both PVC and cornstarch suspensions exhibit discontinuous shear thickening where the gradient $d(\log \eta)/d(\log \sigma)$ reaches 1 (vertical flow curve when plotting $\eta = f(\dot{\gamma})$). Within experimental uncertainty, shear thickening begins at a fixed onset stress, which depends only on the studied system and not on the volume fraction $\phi^2$ (see dotted lines on Supplementary Fig. 3).

For PVC particles in a suspending liquid made of 100 vol.% plasticizer, an onset stress of 75 ± 5 Pa is found. For PVC particles in a suspending liquid made of 90 vol.% of plasticizer an onset of 38 ± 5 Pa is measured and for PVC particles in a

suspending liquid made of 67 vol.% of plasticizer an onset of $8 \pm 2$ Pa is found. For cornstarch particles suspended in pure water, an onset stress of 3–4 Pa in found, in good agreement with previous works[21,22].

**Data availability.** The data that support the findings of this study are available from the corresponding author on request.

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

## Acknowledgements

We thank Cyprien Poirier for fabrication of the PVC substrates and Mario S. Rodrigues for many fruitful discussions. J.C., A.N. and A.S. acknowledge funding from the European Union's H2020 Framework Programme/ERC Starting Grant agreement number 637748—NanoSOFT. L.B. acknowledges support from the European Union's FP7 Framework Programme/ERC Advanced Grant Micromegas and funding from a PSL chair of excellence. J.C., A.N., L.B. and A.S. acknowledge funding from ANR project BlueEnergy.

## Author contributions

J.C., G.C and A.C wrote the paper. J.C. performed the experiments and analyzed the data dealing with AFM. G.C performed the experiments and analyzed the data dealing with rheology. L.B., A.N. and A.S. discussed the experiments. A.C.,L.B,A.S. supervised the project. All authors discussed the results and commented on the manuscript.

## Additional information

**Competing interests:** The authors declare no competing financial interests.

**Publisher's note**: 

