## [Peer Review File · Nature Communications]

REVIEWERS' COMMENTS:

Reviewer #1 (Remarks to the Author):

This is a fascinating and well written paper reporting a breakthrough in our understanding of a ubiquitous phenomenon in a ubiquitous class of materials: the shear thickening of concentrated suspensions, made popular with the public in the 'running on cornstarch' experiment. As the authors correctly stated, there has been uncertainty and controversy in the mechanism for some time, viz., whether it is due to hydrodynamic (lubrication) interaction or frictional contact between particles under stress. Some pretty cogent evidence has been accumulated recently from theory, simulations, and macroscopic rheological experiments that the latter is the correct explanation. This paper adds dramatic evidence in its favour by providing microscopic evidence on the particle level that is referenced back to rheological data. There is little doubt in my mind that the main scientific conclusions are correct, and the advance is substantial enough to be reported in this journal. The following are some comments for the authors to consider in the final revision of their paper to make its impact as high as possible:

Abstract: the correct English is 'long-needed', not 'long-time needed'.

Opening paragraph: discontinuous shear thickening alone is not sufficient to explain 'running on corn starch', which relies on transients. The authors should consider making this clearer and citing relevant work, e.g., from the groups of Brown (propagation of dynamical impact front) and Poon (unsteady flow). Also, in the range of applications the authors mention, they could broaden the appeal even further by mentioning the role of shear thickening in determining the structure of rocks formed by cooling magma (see, e.g., Sumita and Manga, *Earth & Planetary Sci Letters* 269 (2008) 468).

In the third paragraph, the authors claim that elastohydrodynamic forces alone could explain continuous shear thickening, and cite [11] in support. If I remember correctly, that paper is a simulation study, and therefore cannot be considered as having established beyond all reasonable doubt the role of such forces. In science, experiments have to be the final arbiter of truth. As far as I know, there has been little systematic comparison between any version of the hydrodynamic shear thickening theory with experimental data to show that hydrodynamics is adequate for explaining even the full range of observed continuous shear thickening. So perhaps the authors should not give credit where credit is not due.

The final paragraph of the introduction claims that PVC and cornstarch suspensions are 'well known' shear-thickening systems. References should be given here, but also later in discussion of Fig 4 (and associated data in the supplement) to show that the authors here reproduce the well known behaviour.

My main scientific issue with the main part of the paper is to do with distinguishing between sliding and static friction. The authors here measure the sliding friction between particles. A reader unfamiliar with the recent shear-thickening literature may therefore be led to think that it is sliding friction that increases the dissipation and therefore gives rise to shear thickening. However, the current understanding (e.g. see a PNAS by Wyart) is that it is static friction that 'does the trick'. Static friction imposes extra constraints on local motion, so that to achieve the same macroscopic affine strain, the system has to undergo more extensive local non-affine motions when static friction has been 'turned on'. Hence the higher dissipation. So I think the authors should make this clear.

Doing so is more than just a service to the reader. I wonder if the discrepancy between the measured slopes in Fig. 4b (0.006 and 0.02) and the predicted value from simulation (0.05) may be due to the

distinction between static and dynamic friction. This slope relates the macroscopic onset shear stress and the microscopic friction. In the simulations giving 0.05, it is static friction that is causing shear thickening; in these experiments, it is sliding friction that is being characterised. The latter admittedly gives a lower bound to the static friction coefficient, but I do not know if the slope here is simply dependent linearly on the microscopic friction coefficient. Some comment on the static/dynamic friction distinction may be appropriate in this context, too.

This is not a criticism of the basic methodology and finding - I do not see how this method could yield anything else, and it has already yielded a lot by studying sliding friction. But in principle, a method that characterises static friction would bring us even nearer a 'perfect smoking gun'.

Reviewer #2 (Remarks to the Author):

Publication is recommended.

This paper reports a novel investigation of the frictional force between two colloidal particles measured by afm and its correlation (in Figure 4) to the onset stress of shear thickening, which has been attributed to forcing frictional contacts between such particles. As the authors note, their findings are an important bridge, connecting microscopic and macroscopic measurements, Eq. 4. The microscopic friction is here measured to be comparable but several times larger than the prediction from a shear thickening model (ref. 30). What is the effect of roughness?

The experiments are well designed. For example, symmetric excitation of the afm probe is a valuable feature. Please report some other details such as the amplitude of motion (a very broad range 1-50 nm is reported; please give more information. aT is said to be constant.), repeatability of measurements and conditions that cause damage to the probe particle.

On page 2, perhaps qualify the statement that elastohydrodynamics captures continuous shear thickening, because it predicts a broader transition than observed experimentally.

The term "sliding" direction seems inappropriate when $i = N$.

Several recurring typos: Newtonian and Brownian should be capitalized.

Reviewer #3 (Remarks to the Author):

In this manuscript, the authors study experimentally the friction between two colloidal particles, so as to test decisively a recent microscopic theory of shear thickening- one of the most spectacular properties of suspensions. This is a very active field, and this contribution is important: to my knowledge it is the first direct microscopic measurement of frictional properties at a contact level between colloids. This is very precious, and of interest that goes well beyond shear thickening. The paper is well-written, and the experiments appears to have been done with great care (but I am not an experimentalist so other reviewers will hopefully comment more specifically on that).

It rarely happens, but I don't have many comments to improve the manuscripts. It is very convincing, I think it will have a high impact and will please the broad readership of Nature Communications.

REVIEWERS' COMMENTS:

Reviewer #1 (Remarks to the Author):

This is a fascinating and well written paper reporting a breakthrough in our understanding of a ubiquitous phenomenon in a ubiquitous class of materials: the shear thickening of concentrated suspensions, made popular with the public in the 'running on cornstarch' experiment. As the authors correctly stated, there has been uncertainty and controversy in the mechanism for some time, viz., whether it is due to hydrodynamic (lubrication) interaction or frictional contact between particles under stress. Some pretty cogent evidence has been accumulated recently from theory, simulations, and macroscopic rheological experiments that the latter is the correct explanation. This paper adds dramatic evidence in its favour by providing microscopic evidence on the particle level that is referenced back to rheological data. There is little doubt in my mind that the main scientific conclusions are correct, and the advance is substantial enough to be reported in this journal. The following are some comments for the authors to consider in the final revision of their paper to make its impact as high as possible:

We thank the reviewer for his/her positive comments on our manuscript.

Abstract: the correct English is 'long-needed', not 'long-time needed'.

We corrected the typo.

Opening paragraph: discontinuous shear thickening alone is not sufficient to explain 'running on corn starch', which relies on transients. The authors should consider making this clearer and citing relevant work, e.g., from the groups of Brown (propagation of dynamical impact front) and Poon (unsteady flow). Also, in the range of applications the authors mention, they could broaden the appeal even further by mentioning the role of shear thickening in determining the structure of rocks formed by cooling magma (see, e.g., Sumita and Manga, *Earth & Planetary Sci Letters* 269 (2008) 468).

Following the advice of the referee, we mention the role of shear thickening in nature.

We write :

Flows of concentrated dispersions are ubiquitous in nature and industry: water or oil saturated sediments, muds, crystal-bearing magma \cite{sumita2008suspension}, concrete, silica suspensions, cornflour mixtures, latex suspensions and clays are example of shear thickening systems.

We cite relevant work dealing with explanation of 'running on corn starch' and explain this point in more details.

We write: Under impact, the formation of a dynamic jamming front makes the shear thickening fluid so resistant that a person can run on it \cite{van2012soft,waitukaitis2012impact,mukhopadhyay2014shear}

In the third paragraph, the authors claim that elasto-hydrodynamic forces alone could explain continuous shear thickening, and cite [11] in support. If I remember correctly, that paper is a simulation study, and therefore cannot be considered as having established beyond all reasonable doubt the role of such forces. In science, experiments have to be the final arbiter of truth. As far as I know, there has been little systematic comparison between any version of the hydrodynamic shear thickening theory with experimental data to show that hydrodynamics is adequate for explaining even the full range of observed continuous shear thickening. So perhaps the authors should not give credit where credit is not due.

We tempered our description from the work and write:

"When normal elasto-hydrodynamic contact forces are taken into account (without solid friction), these simulations capture continuous shear thickening and large increase in suspension viscosity [11]. However, this model predicts shear rate independent rheology for non-Brownian systems and broader transition than observed experimentally."

The final paragraph of the introduction claims that PVC and cornstarch suspensions are 'well known' shear-thickening systems. References should be given here, but also later in discussion of Fig 4 (and associated data in the supplement) to show that the authors here reproduce the well known behaviour.

We give the references in introduction:

We study two well known shear thickening systems: polyvinyl chloride (PVC) suspended in various solvents \cite{Willey1978, Willey1982} and cornstarch particles suspended in water \cite{fall2015macroscopic, Hermes2016}.

We added the references for both systems in the intro as well as in the discussion of Fig. 4.

My main scientific issue with the main part of the paper is to do with distinguishing between sliding and static friction. The authors here measure the sliding friction between particles. A reader unfamiliar with the recent shear-thickening literature may therefore be led to think that it is sliding friction that increases the dissipation and therefore gives rise to shear thickening. However, the current understanding (e.g. see a PNAS by Wyart) is that it is static friction that 'does the trick'. Static friction imposes extra constraints on local motion, so that to achieve the same macroscopic affine strain, the system has to undergo more extensive local non-affine motions when static friction has been 'turned on'. Hence the higher dissipation. So I think the authors should make this clear. Doing so is more than just a service to the reader. I wonder if the discrepancy between the measured slopes in Fig. 4b (0.006 and 0.02) and the predicted value from simulation (0.05) may be due to the distinction between static and dynamic friction. This slope relates the macroscopic onset shear stress and the microscopic friction. In the simulations giving 0.05, it is static friction that is causing shear thickening; in these experiments, it is sliding friction that is being characterised. The latter admittedly gives a lower bound to the static friction coefficient, but I do not know if the slope here is simply dependent linearly on the microscopic friction coefficient. Some comment on the static/dynamic friction distinction may be appropriate in this context, too. This not a criticism of the basic methodology and finding - I do not see how this method could yield anything else, and it has already yielded a lot by studying sliding friction. But in principle, a method that characterises static friction would bring us even nearer a 'perfect smoking gun'.

We agree with the referee, we measure slidding friction We added a note at the beginning of Page 5 :
“Note that we can not measure static friction forces with our set-up. However, we expect static friction to appear concurrently to sliding dynamic friction.”

We also added a note after Eq. (4):

“We find here a proportionnality coefficient $\beta_{\text{PVC}} \approx 0.006$ and $\beta_{\text{Cornstarch}} \approx 0.02$, in relatively good agreement with predictions from simulations performed on smooth particles ($\beta_{\text{simu}} \approx 0.05$ for a friction coefficient $\mu=1$ \cite{mari2014shear}). Those coefficients characterize stress transmission from the suspension to the particles level and depend on the microscopic friction coefficient \cite{mari2014shear} for both static and sliding friction, as well as particle shape and roughness. Note that in the simulations, the values of the static and dynamic friction are assumed to be the same, which may not be the case in our situation. Roughness may also block the particles and increase the value of β found by the numerical simulations”

Reviewer #2 (Remarks to the Author):

Publication is recommended.

This paper reports a novel investigation of the frictional force between two colloidal particles measured by afm and its correlation (in Figure 4) to the onset stress of shear thickening, which has been attributed to forcing frictional contacts between such particles. As the authors note, their findings are an important

bridge, connecting microscopic and macroscopic measurements, Eq. 4. The microscopic friction is here measured to be comparable but several times larger than the prediction from a shear thickening model (ref. 30). What is the effect of roughness?

Particle roughness is characterized by AFM topography scans, as shown in supplementary figures 1 and 2. Local particle roughness will have an effect on the value of the friction coefficient, which we can measure through out set-up. More macroscopic roughness may also lead to changes of the prefactor β compared to current results from simulations, for example by preventing rolling between particles. Other causes of disagreement might relate to the fact that numerical simulations made no differences between static and dynamic friction. In this work we could characterize only dynamic friction.

We write :

“We find here a proportionality coefficient $\beta_{\text{PVC}} \approx 0.006$ and $\beta_{\text{Cornstarch}} \approx 0.02$, in relatively good agreement with predictions from simulations performed on smooth particles ($\beta_{\text{simu}} \approx 0.05$ for a friction coefficient $\mu=1$ \cite{mari2014shear}). Those coefficients characterize stress transmission from the suspension to the particles level and depend on the microscopic friction coefficient \cite{mari2014shear} for both static and sliding friction, as well as particle shape and roughness. Note that in the simulations, the values of the static and dynamic friction are assumed to be the same, which may not be the case in our situation. Roughness may also block the particles and increase the value of β found by the numerical simulations.”

The experiments are well designed. For example, symmetric excitation of the afm probe is a valuable feature. Please report some other details such as the amplitude of motion (a very broad range 1-50 nm is reported; please give more information. aT is said to be constant.), repeatability of measurements and conditions that cause damage to the probe particle.

aT is kept constant by a feed-back loop during one measurement but its value can be changed to test the velocity dependence of the friction coefficient.

We added in the Methods :

The absolute value of a_T can be varied from 10 to 60 nm to test the dependence of the friction coefficient on speed (Fig. 3b). A fixed excitation voltage E_{N} is applied to the normal mode, and dissipation is measured by monitoring the oscillation amplitude a_{N} , of the order or less than 1 nm.

On page 2, perhaps qualify the statement that elasto-hydrodynamics captures continuous shear thickening, because it predicts a broader transition than observed experimentally.

We added a note on Page 2:

“When normal elasto-hydrodynamic contact forces are taken into account (without solid friction), these simulations capture continuous shear thickening and large increase in suspension viscosity [11]. However, this model predicts shear rate independent rheology for non-Brownian systems and broader transition than observed experimentally.”

The term “sliding” direction seems inappropriate when $i = N$.

We clarified:

“We can then express the frictional forces applied on the bead as the sum of viscous-like friction force (proportional to speed) and solid friction force (independent of speed and corresponding to sliding friction for tangential motion, $i = T$)”

Several recurring typos: Newtonian and Brownian should be capitalized.

We capitalized those words.

Reviewer #3 (Remarks to the Author):

In this manuscript, the authors study experimentally the friction between two colloidal particles, so as to test decisively a recent microscopic theory of shear thickening- one of the most spectacular properties of suspensions. This is a very active field, and this contribution is important: to my knowledge it is the first direct microscopic measurement of frictional properties at a contact level between colloids. This is very precious, and of interest that goes well beyond shear thickening. The paper is well-written, and the experiments appears to have been done with great care (but I am not an experimentalist so other reviewers will hopefully comment more specifically on that). It rarely happens, but I don't have many comments to improve the manuscripts. It is very convincing, I think it will have a high impact and will please the broad readership of Nature Communications.

We thank the referee for his/her valuable comments.